# In-Context Neural PDE: learning to adapt a neural solver to different physics

## Abstract

We address the problem of predicting the next state of a dynamical system governed by *unknown* temporal partial differential equations (PDEs) using limited time-lapse data. While transformers offer a natural solution to this task through in-context learning, the inductive bias of temporal PDEs suggests a more tailored and effective approach. Specifically, when the underlying temporal PDE is fully known, classical numerical solvers can evolve the state with only a few parameters. Building on this observation, we introduce a large transformer-based hypernetwork that processes successive states to generate parameters for a much smaller neural ODE-like solver, which then predicts the next state through time integration. This framework, termed as *in-context neural PDE*, decouples parameter estimation from state prediction, offering closer alignment with classical numerical methods for improved interpretability while preserving the in-context learning capabilities of transformers. Numerical experiments on diverse physical datasets demonstrate that our method outperforms standard transformer-based models, reducing sample complexity and improving generalization, making it an efficient and scalable approach for spatiotemporal prediction in complex physical systems.

## 1 Introduction

Modeling dynamical systems from data is a highly active area of research with the potential to significantly reduce both computational costs (Kidger, 2021) and the need for ad-hoc engineering to predict future states (Farlow, 2012). In this paper, we consider dynamical systems described by partial differential equations (PDEs). The conventional data-driven approach typically involves "learning" a fixed dynamic model from a large number of trajectories obtained from various initializations, which is often referred to as neural operator learning (Kovachki et al., 2023; Boullé & Townsend, 2023). However, in many practical scenarios, the exact governing physical laws are unknown and may vary across different trajectories, making those methods unsuitable and necessitating a more flexible data-driven approach. To address this limitation, we investigate the problem of predicting the next state of a system from a few successive time-lapse observations, where the underlying dynamics are unknown and can vary between different trajectories. This problem is more complex as it implicitly requires both estimating the underlying physics of the dynamical system (inverse problem) and integrating it in time (forward problem). In contrast, these two tasks are typically tackled independently in existing literature (Blechschmidt & Ernst, 2021).

One promising way to handle scenarios with unknown or varying dynamics is through models that can effectively capture context—a sequence of preceding states—to infer the temporal patterns governing the system. Transformers, known for their strong sequential learning capabilities (Vaswani et al., 2017), excel in this area through a mechanism called In-Context Learning (ICL) (Brown et al., 2020), allowing them to adapt flexibly to different dynamics using prior state information. This adaptability has enabled transformers to achieve remarkable performance in natural language processing, where context plays a crucial role (Dubey et al., 2024). Recent works (McCabe et al., 2024; Yang & Osher, 2024; Yang et al., 2023; Liu et al., 2023) have also shown that transformers, when trained on diverse PDE dynamics and initial conditions, can predict future states across different contexts, positioning them as powerful tools for data-driven modeling of dynamical systems in physical domains.

However, applying transformers to physical systems still remains challenging, as they often require vast amounts of data to avoid overfitting and can struggle with predicting out-of-distribution trajectories, leading to instabilities over multi-step rollouts (McCabe et al., 2023; 2024). In contrast, classical numerical methods excel when the governing dynamics are known, evolving the system accurately with minimal parameters and without any data training. These methods also often preserve important structural properties of physics, such as continuous-time evolution and translation equivariance (Mallat, 1999), which transformers do not naturally inherit. This lack of structure preservation contributes to the limitations of transformers in physical applications, particularly in terms of high sample complexity and instability.

In this work, we propose a novel framework named *in-context neural PDE* (IC-NPDE) that combines the best of both worlds: an estimate of the parameters through context is incorporated into a neural ODE-like solver (Chen et al., 2018). Specifically, a large hypernetwork processes successive states of a trajectory to generate parameters for a much smaller neural PDE solver, which then predicts the next state through time integration. In this manner, we *decouple* the parameter estimation of the dynamical context from state prediction, unlike previous transformer-based methods that tackle both tasks jointly. Our decoupling approach can be viewed as a meta-learning algorithm (Thrun & Pratt, 1998; Finn et al., 2017), a class of algorithms designed to self-adapt to specific tasks. In our setting, predicting the next state from limited time-series data is treated as a task that depends on (1) initial conditions, (2) physical parameters, and (3) even different underlying physical laws. The neural network used in the ODE-like solver respects the underlying continuous-time nature of the physics and preserves spatial translation equivariance through the use of Convolutional Neural Networks (CNNs). The parameters are fully learnable, offering greater flexibility in estimating spatial derivatives (Bar-Sinai et al., 2019). Furthermore, our integrated network uses far fewer parameters compared to a transformer, creating an information bottleneck (Tishby et al., 2000). This bottleneck forces the system to focus on the most essential aspects of the dynamics, preventing overfitting and improving generalization, particularly for out-of-distribution states. By concentrating the representation in a smaller parameter space, the model gains robustness and efficiency, making it more suitable for predicting complex physical systems.

**Contributions.** Our key contributions are as follows: **(a)** To our knowledge, this is the first work to combine an ICL approach with differentiable PDE solvers for spatiotemporal prediction. **(b)** We propose a framework that introduces a tailored inductive bias for physical systems, providing improved interpretability by aligning more closely with classical numerical methods. **(c)** Our method demonstrates superior learning performance across multiple physical systems using standard datasets in the literature and **(d)** achieves better numerical accuracy on multi-step rollouts compared to state-of-the-art ICL models, with improved generalization. Upon publication, we will open-source our implementation.

## 2 RELATED WORKS

**Classical and neural solvers.** While classical PDE solvers remain the state-of-the-art for achieving high precision, neural network-based surrogate solvers (Lu et al., 2019; Li et al., 2020; Kovachki et al., 2023) have opened up new possibilities for inferring approximate solutions quickly for certain PDEs. However, these solvers need to be trained on samples derived from the same PDE. Some variants, such as (Karniadakis et al., 2021; Kochkov et al., 2021), allow the incorporation of corrective terms to approximate trajectory dynamics, but they still lack adaptability to the context, which is a key feature of our method. Symbolic regression (Lemos et al., 2023) separates trajectory parameter inference from the integration task but struggles to scale to high-dimensional data and handle large search spaces effectively. In contrast, our method directly approximates the differential operators using a cascade of small convolutional layers, enabling efficient learning in high-dimensional data.

**Meta-learning strategies applied to dynamical systems.** As an inspiration for our work, Metz et al. (2022) pretrains an optimizer on a large collection of datasets and models, which can be viewed as meta-learning a specific discretized dynamical system in order to surpass stochastic gradient descent. In a different setting, Gusak et al. (2021); Guo et al. (2022) propose strategies to learn the best ODE solver from a family of solvers, which could be combined with our approach. Meanwhile, Bar-Sinai et al. (2019) focuses on meta-learning differentiable filters for PDEs, aligning with one

aspect of our methodology. In a different setting, de Avila Belbute-Peres et al. (2021) learns to map PDE parameters to the corresponding physics-informed neural netowrk's parameters (Karniadakis et al., 2021); however, this lacks capability for ICL. The most closely related works to ours, to the best of our knowledge, are McCabe et al. (2024); Yang et al. (2023); Liu et al. (2023); Yang & Osher (2024), which utilize transformers for in-context learning of neural operators for PDEs. However, as discussed in the introduction, these methods lack inductive bias for physics and tend to require a large amount of data to generalize. Xian et al. (2021); Wang et al. (2022) use an encoder/decoder strategy to embed dynamics in a latent space for predicting the next state; while related, these approaches are conceptually closer to McCabe et al. (2024).

## 3 METHODOLOGY

### 3.1 PRELIMINARIES

**Problem setting and notations.** Let $t$ denote time and $x \in \mathbb{R}^n$ be the spatial variable. We consider a class of PDEs of spatial order $k$ for function $u : \mathbb{R} \times \mathbb{R}^n \to \mathbb{R}^m$ of the form

$$\partial_t u_t(x) = g\big(u_t(x),\, \partial_{x_1} u_t(x),\, \partial_{x_2} u_t(x),\, \ldots,\, \partial^2_{x_1 x_1} u_t(x),\, \partial^2_{x_1 x_2} u_t(x) \ldots\big), \qquad (1)$$

where different $g$ corresponds to different physics (different equations and/or parameters) governing the system. Here, $n$ is the spatial dimension, typically ranging from 1 to 3 in physical systems, and $m$ is the number of physical variables described by the PDE. For notational simplicity, we assume the PDE is time-homogeneous (i.e., $g$ does not depend on $t$) though our methods can be straightforwardly extended to the time-inhomogeneous case. For PDEs with higher-order time derivatives, specifically of order $r$, we can redefine the function of interest as $\bar{u} = (u, \partial_t u, \ldots, \partial^r_t u)$, which allows us to rewrite the PDE in the form of Eq. 1. Given a spatiotemporal trajectory with *unknown* $g$, our goal is to predict the state $u_{t+1}$ using the preceding $T$ successive states $u_{t-T+1}, \ldots, u_t$, where $T$ is referred to as the *context length*. In practice, the observed $u_t$ is discretized over grid points (assumed fixed and uniformly-distributed in this paper) rather than being a continuous spatial function. For simplicity, we use continuous notation throughout.

**Finite difference and neural PDEs.** Finite difference method is a classical approach for numerically solving an explicitly given PDE by discretizing spatial derivatives on a grid. This connection forms a basis for leveraging CNNs to approximate PDE solutions. To illustrate this relationship, consider the standard 2D diffusion equation:

$$\partial_t u_t(x) = \beta \Delta u_t(x),\ x \in [0,1]^2, \qquad (2)$$

subject to periodic boundary conditions, where $\beta > 0$ is the diffusion coefficient. Let the state $u_t$ be discretized over a uniform grid with spacing $\Delta x$, denoted by $u_{t,i,j}$ for $0 \le i, j \le N$. Using a standard centered finite difference scheme, the right-hand side of Eq. 2 can be approximated as:

$$\partial_t u_{t,i,j} \approx \beta \frac{u_{t,i+1,j} + u_{t,i-1,j} + u_{t,i,j+1} + u_{t,i,j-1} - 4u_{t,i,j}}{(\Delta x)^2} = u_{t,i,j} \star \theta,$$

where $\theta = \beta/(\Delta x)^2 [0, 1, 0; 1, -4, 1; 0, 1, 0]$ is a small $3 \times 3$ convolution filter, and $\star$ denotes the convolution operation with periodic padding. Defining $\tilde{f}_\theta(u) := u \star \theta$, we have

$$u_{t+1} \approx u_t + \int_t^{t+1} \tilde{f}_\theta(u_s)\, ds,$$

which indicates that when the physics is explicitly known (Eq. 2), we only need a single convolution layer with a few parameters to evolve the solution $u_t$ accurately. Extending this concept to machine learning, Bar-Sinai et al. (2019) proposed replacing these fixed convolutional coefficients with learnable parameters, resulting in improved accuracy on certain grid sizes. For more complex PDEs involving nonlinear dynamics, additional layers with nonlinear activation functions are necessary to capture the underlying effects. This connection between finite difference methods and CNNs highlights that relatively small CNN architectures, compared to transformer-based models, can effectively represent temporal PDEs while preserving spatial translation equivariance. This perspective has been explored in previous work (Ruthotto & Haber, 2020) and recently applied to design neural architectures for solving PDEs when the governing physics is known (Liu et al., 2024).

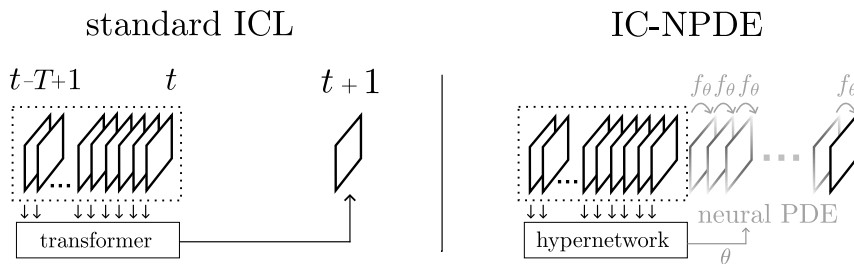

Figure 1: Illustration of our method. Unlike a standard ICL approach based on a transformer such as AViT, which predicts the next frame $u_{t+1}$ directly from the context $(u_{t-T+1}, \dots, u_t)$, our in-context neural PDE (IC-NPDE) framework first uses a hypernetwork to predict the parameters $\theta$ for a smaller neural PDE solver. This solver integrates a CNN $f_\theta$ in time to predict the next state.

We note that the above connection holds primarily for continuous-time or small discrete-time steps. For larger time steps, the exact solution to Eq. 2 still retains a convolutional structure: $u_{t+\Delta t}(x) = \int_{\mathbb{R}^2} u_t(x-y)G(\Delta t, y)\, dy$ where $G(\Delta t, y) = (4\pi\beta\Delta t)^{-1}\exp(-\|y\|^2/(4\beta\Delta t))$. However, for large $\Delta t$ (specifically when $\beta\Delta t \gg 1$), the convolution kernel $G(\Delta t, y)$ decays slowly, resulting in a significantly larger effective receptive field. Consequently, using a CNN $\tilde{f}_\theta$ to directly approximate the solution in the form of $u_{t+\Delta t}(x) = \tilde{f}_\theta(u_t)$, as proposed in (Wang et al., 2022), would require a much larger network with more parameters, making the optimization more challenging.

**Transformers for PDEs.** Recent works (McCabe et al., 2024; Yang & Osher, 2024; Yang et al., 2023; Liu et al., 2023) have proposed to use ICL predictors based on transformers to directly estimate the next state from a few successive iterates, in other words

$$\hat{u}_{t+1} = F_\theta(u_{t-T+1}, \dots, u_t),\tag{3}$$

where typically $F_\theta$ is an over-parameterized transformer or diffusion model. These models can work without knowing the physics a priori, in contrast to the common neural PDE solvers. However such models are more prone to over-fitting as they do not incorporate good inductive biases, such as explicit differential operators, contrary to Eq. 3.

### 3.2 METHOD: IN-CONTEXT NEURAL PDE

**Our framework.** In this work, we propose a framework that combines the best of both worlds from ICL and classical numerical schemes (see Fig. 1): the parameters $\theta \in \mathbb{R}^{d_1}$ of a small convolutional neural network $f_\theta$—referred to as the *integrated network*—are generated from an ad-hoc model $\psi_\alpha$, referred to as a *hypernetwork* (Ha et al., 2016), which uses the context to estimate these parameters, leading to the formal equations:

$$\begin{cases} \hat{u}_{t+1} = u_t + \displaystyle\int_t^{t+1} f_\theta(u_s)\, ds\,, \\ \theta = \psi_\alpha\big(u_{t-T+1}, \dots, u_t\big)\,, \end{cases}\tag{4}$$

where $\alpha \in \mathbb{R}^{d_2}$. This formulation significantly structures the predictor, which now has a convolutional structure aligned with Eq. 1: the spatial derivatives are necessarily approximated using the convolutional kernels of $f_\theta$.

By leveraging auto-differentiation and the strategy of Chen et al. (2018), the above two equations can be learned jointly in an end-to-end manner by solving the optimization problem

$$\min_\alpha \frac{1}{|\mathcal{D}|} \sum_{\ell \in \mathcal{D}} \text{Loss}(u_{t+1}^\ell, \hat{u}_{t+1}^\ell)\,,$$

where each $\ell$ in the dataset $\mathcal{D}$ is a data point in the form of $(u_{t-T+1}^\ell, \dots, u_t^\ell, u_{t+1}^\ell)$. In this paper, we choose the loss function as the normalized root mean square error (NRMSE); see Appendix C for the detailed definition.

After the weights $\alpha$ are trained, for predicting specific dynamics from a given trajectory as the downstream task, the entire inference process can be viewed as a meta-learning or an ICL procedure, as no hyper-network retraining is involved.

**Hypernetwork design.** Multiple choices of hypernetwork could be used, but we decided to exploit transformers due to their favorable auto-regressive properties (Lin et al., 2022). In fact, we emphasize that any type of meta learning approaches, like directly optimizing $\theta$ via gradient descent, could be considered yet it might not be computationally favorable. Here, we have $d_1 \ll d_2$, in order to limit overfitting and any memorization phenomenon by forcing the transformer to estimate the parameters of the trajectory rather than the next state.

**Numerical integration.** The integral from $t$ to $t+1$ in Eq. 4 is discretized using a fourth-order Runge-Kutta method (Kassam & Trefethen, 2005). Gradient backpropagation is performed using an adjoint sensitivity method which scales linearly in the number of integration steps and has low memory cost (Chen et al., 2018). The solver uses 30 integration steps, but ablation studies show that this number is quite conservative, as explained in Tab. 4. One could either reduce this number or use the same number but to predict further into the future. For example, for slowly varying fluids, one may want to predict up to $t+2$ in a single forward pass with 15 integration steps for each interval $[t, t+1], [t+1, t+2]$. We leave this possibility for future work.

## 4 NUMERICAL EXPERIMENTS

### 4.1 GENERIC ARCHITECTURES

**Integrated network architecture.** The integrated network $f_\theta$ is a CNN starting with a $k \times k$ convolution, acting as a local differential operator, followed by six $1 \times 1$ convolutions that apply a pointwise function. In detail, Let $C_{\text{in}}$ and $C_{\text{hidden}}$ represent input and hidden channels, respectively, with $C_{\text{out}} = C_{\text{in}}$. The first convolution has weights of shape $(C_{\text{hidden}}/2, C_{\text{in}}, k, k)$ and includes a skip connection via concatenation. This is followed by a $(C_{\text{hidden}}, C_{\text{in}} + C_{\text{hidden}}/2, 1, 1)$ convolution. Next, two blocks are applied, each consisting of two convolutions with weights $(2C_{\text{hidden}}, C_{\text{hidden}}, 1, 1)$ and $(C_{\text{hidden}}, 2C_{\text{hidden}}, 1, 1)$, followed by a skip connection. The final layer uses weights of shape $(C_{\text{out}}, C_{\text{hidden}}, 1, 1)$. Each $1 \times 1$ convolution is followed by group normalization (8 groups) and a GeLU activation. For datasets with non-periodic boundary conditions, we use zero-padding for the spatial convolution and we add manually a mask of the boundaries as input to $f_\theta$.

**Hypernetwork architecture.** In all our experiments we assume $\psi$ in Eq. 4 is a transformer. It as a CNN encoder and a MLP decoder outputting the parameters. Its encoder consists of three convolution layers of kernel sizes of $4, 2, 2$ respectively, with GeLU activation, ending up with a patch size of 16. The hidden dimension (token space) is 384. After the encoder, we cascade 12 time-space attention blocks, each containing a time attention, and axial attentions along each space dimension (McCabe et al., 2024). Each attention block contains 6 heads and uses relative positional encodings. The output of the attentions is averaged over both time and space, leaving only the channel dimension. Finally, we branch a MLP with two hidden layers that progressively increases the channel dimension to recover the expected parameter shape of $\theta$.

**Multiple physics training.** When jointly trained on multiple datasets, most of the weights in the hypernetwork are shared across datasets, except for the first $1 \times 1$ convolution, which is learned separately for each dataset to accommodate varying channel numbers from input, and the final MLP weights, which are also learned independently per dataset to produce the parameters $\theta$. Regarding the CNN $f_\theta$, $C_{\text{hidden}}$ remains the same across different dataset, but the number of input and output channels in the first and last convolution layers varies depending on the number of channels in the data. Also, the kernel size of the first (and only) spatial filtering in $f_\theta$, is set to $k = 5$ for all the datasets, except for compressible Navier-Stokes where increasing the kernel size to 11 was necessary to obtain good performances. This is certainly due to the fact that the trajectory in this dataset makes much bigger movement from one step to the other.

Table 1: Specifics of the datasets considered in this paper.

| Dimension | Dataset Name | Resolution (pixels) | Sequence Length | Boundary Conditions |
|---|---|---|---|---|
| 1D | Burgers | 1024 | 200 | periodic |
| 2D | Shallow water | $128 \times 128$ | 100 | open |
| 2D | Diffusion-reaction | $128 \times 128$ | 100 | Neumann |
| 2D | Navier-Stokes (incomp.) | $128 \times 128$ | 1000 | Dirichlet |
| 2D | Navier-Stokes (comp.) | $128 \times 128$ | 21 | periodic |
| 2D | Shearflow (incomp.) | $128 \times 256$ | 200 | periodic |
| 2D | Euler (comp.) | $128 \times 128$ | 200 | periodic |

## 4.2 LARGE SCALE EXPERIMENTS, PERFORMANCES AND SAMPLE EFFICIENCY

**Datasets description.** In order to explore a variety of physics, we consider seven datasets. One dataset consists of 1D signals, while the other six contain 2D signals. Each dataset includes simulations of a specific PDE, with potentially varying coefficients, and trajectories evolved from different initial conditions. Detailed descriptions of the datasets are provided in Tab. 1. Burgers, shallow water, diffusion reaction, Navier-Stokes (incompressible and compressible) datasets are sourced from PDEBench (Takamoto et al., 2022). The two additional datasets include a 2D periodic incompressible shearflow, generated using the `Dedalus` software (Burns et al., 2020), and a dataset based on Euler equations—a special case of compressible Navier-Stokes equations—produced using the `CLAWPack` software (Mandli et al., 2016; Clawpack Development Team, 2021). To allow for computation of the models on all the datasets, we subsampled each data to a resolution of $128 \times 128$ pixels, except for the Burgers equations data that is 1024 long and the shearflow data that is of resolution $128 \times 256$. Details on the equations, initial conditions, boundary conditions, and data generation can be found in Appendix A.

With the architectures described in Sec. 4.1, we train our IC-NPDE model on multiple datasets and compare it with the Axial Vision Transformer (AViT) (McCabe et al., 2024), which is designed for ICL of multiple physics. Both models are trained on the first five datasets from PDEBench, consistent with those used in (McCabe et al., 2024). We refer the reader to Appendix C for training details.

Table 2: Next steps prediction performances for models trained jointly on multiple datasets.

| Test Dataset | Model | NRMSE | | | | |
|---|---|---|---|---|---|---|
| | | $t+1$ | $t+4$ | $t+8$ | $t+16$ | $t+32$ |
| Burgers | AViT | 0.013 | 0.048 | 0.11 | 0.19 | 0.40 |
| | IC-NPDE (ours) | **0.0036** | **0.022** | **0.082** | **0.11** | **0.37** |
| Shallow-water | AViT | 0.0016 | 0.0097 | **0.032** | **0.033** | 0.098 |
| | IC-NPDE (ours) | **0.00017** | **0.0087** | **0.032** | 0.039 | **0.089** |
| Diffusion-reaction | AViT | 0.012 | 0.12 | 0.35 | 0.47 | **0.76** |
| | IC-NPDE (ours) | **0.00060** | **0.11** | **0.34** | **0.46** | **0.76** |
| Navier-Stokes (incomp.) | AViT | 0.024 | 0.054 | 0.10 | 0.24 | 0.57 |
| | IC-NPDE (ours) | **0.0042** | **0.021** | **0.054** | **0.088** | **0.19** |

Tab. 2 shows the NRMSE over different datasets the two models are trained on (Navier-Stokes (comp.) is excluded from multi-step testing due to its short sequence length). As shown, our model outperforms AViT on next state prediction for all the datasets presented. Furthermore, the accuracy of the rolled-out trajectories are also improved in most of the cases, see Fig. 4 for typical examples. Note that this is achieved with more than three times less learnable parameters than a transformer – 158M compared to 55M in our method – which emphasizes the benefits of incorporating the good inductive bias through a neural PDE in our model.

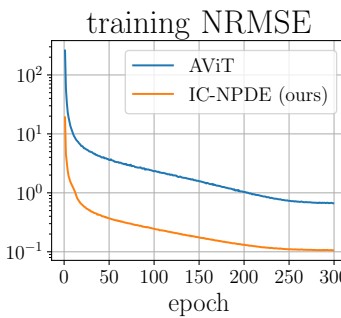 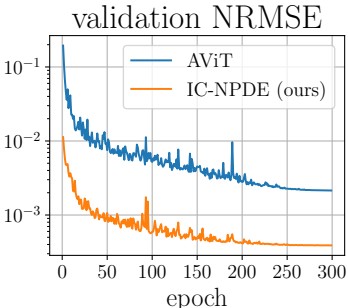

Figure 2: Learning curves on the diffusion-reaction dataset. IC-NPDE already achieves a $10^{-2}$ validation accuracy after one epoch, while a transformer needs around 50 epochs to reach such accuracy. This is an illustration of the better inductive bias implemented in our framework.

This is confirmed by the learning curves showed in Fig. 2. Our model trained on the diffusion-reaction dataset, achieves a good accuracy of $\approx 1\%$ early in the training, after only one epoch, whereas the baseline AViT, which applies a transformer directly to predict the next state, requires 50 epochs to reach the same level. This improved sample efficiency, observed on all the datasets we tested our model on, is also indicative of the benefits of incorporating the good inductive bias in our framework.

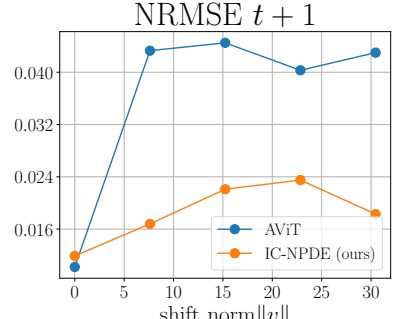

As a concrete example of the issues encountered by a transformer-based architecture, which does not encode translation-equivariance, let us consider shifting all the states in a context, as well as the following state, by a vector $v$ of increasing norm $\|v\|$. Fig. 3 shows that the transformer architecture quickly struggles to predict the next state, although it is competitive when no translation is applied, $\|v\| = 0$.

Figure 3: Performance on the shearflow dataset when context and target are shifted by $v \in \mathbb{R}^2$. While the transformer performs well at $v = 0$, it declines more than IC-NPDE under shifts due to the lack of an inductive bias for translation-equivariance.

### 4.3 INFORMATION BOTTLENECK AND GENERALIZATION PROPERTIES

Our in-context neural PDE model employs a relatively small number of parameters $\theta$ (see Eq. 4) to predict the next state through the integration of $f_\theta$, compared to the typical input sizes. We study the impact of this information bottleneck.

**Parameter space visualization.** When trained on multiple datasets, the parameters $\theta$ returned by our hypernetwork depend on both the initial conditions and the PDE governing the given context. We show that our framework is capable of reducing the variability introduced by the initial condition, allowing it to focus primarily on the PDE dynamics. First, let us separate $\theta = [\theta_{\text{spatial}}, \theta_{\text{pointwise}}]$ into the parameters of the first spatial convolution, and the parameters of the pointwise function that applies to the filtered states. Recall that the filtered states $u \star \theta_{\text{spatial}}$ are analogous to discretizations of the local differential operators such as the gradient and Laplacian in standard numerical solver. The pointwise function parameterized by $\theta_{\text{pointwise}}$, which recombines the filtered states, is driven by the PDE and should contain information about the PDE's coefficients or the PDE itself. We perform dimensionality reduction via Umap (McInnes et al., 2018) over the $32\,768$ parameters $\theta_{\text{pointwise}}$ for $128$ contexts from compressible Navier-Stokes dataset, with two different shear viscosity $\eta = 0.01$ and $\eta = 0.1$. Fig. 5 visualizes the parameter space through these weights $\theta_{\text{pointwise}}$ across different stages of the training, showing that our model progressively identifies two distinct clusters, corresponding to the two physical parameter values. Thus, the hypernetwork clusters contexts that are originated from the same PDE (with same coefficient), despite different initial conditions, which is the key for generalizing to initial conditions.

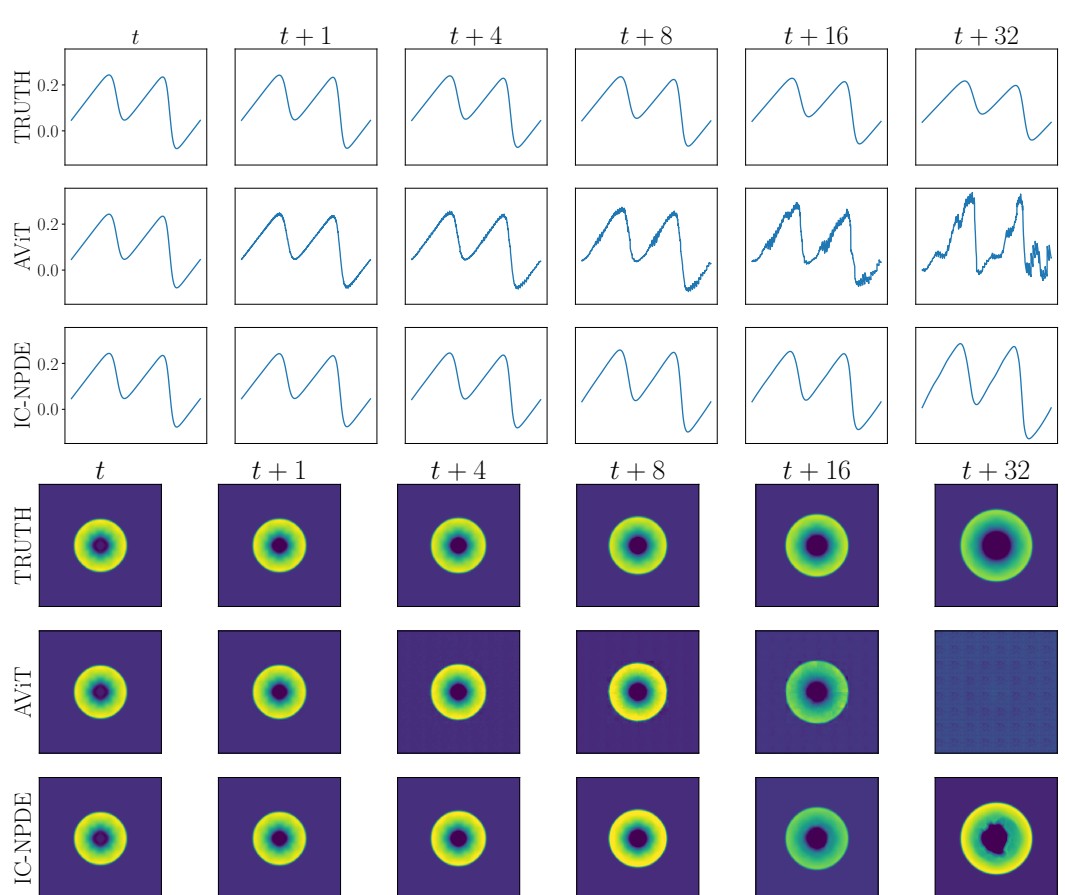

Figure 4: Examples of rollout trajectories from Burgers and shallow-water datasets for models trained jointly on multiple datasets. IC-NPDE leads to more consistent and neat results compared to AViT. See Appendix D for more examples.

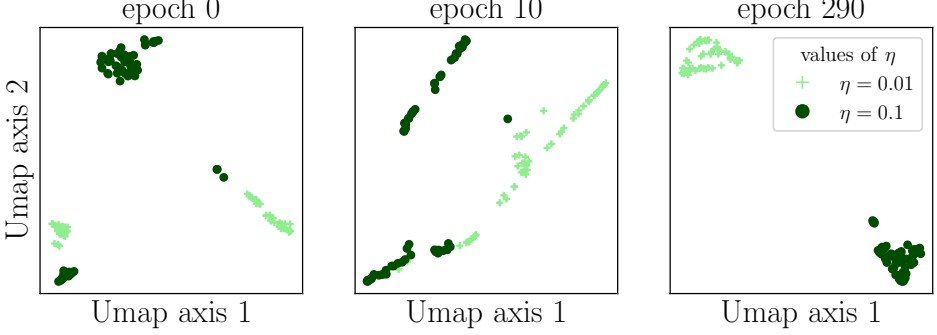

Figure 5: Visualization of the parameter space using UMAP on compressible Navier-Stokes data. Each point represents a set of parameters $\theta$ predicted by the hypernetwork $\psi$ for a given context. The hypernetwork tends to predict similar $\theta$ values when the context is derived from PDEs with the same parameters $\eta$, but different initial conditions. This demonstrates the generalization capability of our approach to varying initial conditions.

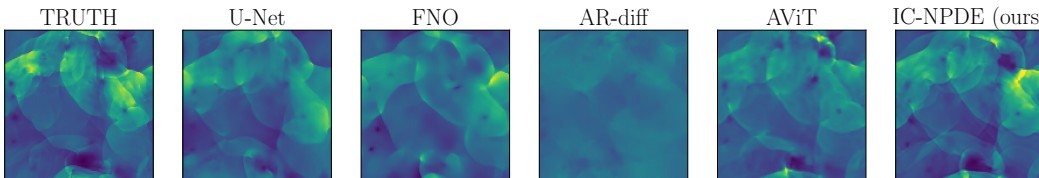

Figure 6: Example of prediction of step $t + 16$ for different models on the Euler dataset. The first three models (U-net, FNO, AR-diffusion) are trained from scratch while the AViT and our IC-NPDE model are finetuned. See Tab. 3 for averaged results.

**Fine-tuning on unseen physics.** We further assess the generalization properties of our IC-NPDE model to an unseen PDE with unseen initial conditions. Specifically, we fine-tune our model and AViT, both initially trained on the previous five datasets, on the Euler dataset. This unseen dataset is governed by compressible Navier-Stokes equations with a single set of parameters and contains initial conditions not encountered during pretraining. Since this dataset is governed by a fixed PDE, we also compare against networks such as U-Net (Ronneberger et al., 2015), Fourier Neural Operators (FNO)(Li et al., 2020), and autoregressive-diffusion models(Kohl et al., 2024), which are designed to learn a fixed operator from data. As shown in Tab. 3 and Fig. 6, our model achieves the best performance when fine-tuned on this new dataset, demonstrating its ability to generalize to a new PDE and novel initial conditions better than the transformer (due to stronger inductive biases) and other neural operator methods (due to the knowledge gained from multiple physics pretraining).

Table 3: Performances of pretrained models, AViT and IC-NPDE (ours), fine-tuned on the unseen Euler equations dataset with a single, fixed, set of coefficients ($\gamma = 1.4$) after 50 epochs. For comparison, we show the performances of 3 other models: Unet, FNO, and auto-regressive diffusion.

| Model | NRMSE | | | | |
|---|---|---|---|---|---|
| | $t + 1$ | $t + 4$ | $t + 8$ | $t + 16$ | $t + 32$ |
| Unet | 0.073 | 0.21 | 0.34 | 0.53 | 0.72 |
| FNO | 0.11 | 0.22 | 0.31 | 0.43 | 0.62 |
| AR-diffusion | 0.13 | 0.27 | 0.38 | 0.48 | **0.53** |
| AViT | 0.067 | 0.13 | 0.30 | 0.40 | 0.84 |
| IC-NPDE (ours) | **0.057** | **0.12** | **0.28** | **0.37** | 0.82 |

### 4.4 Ablation experiments

**Number of integration steps.** The number of steps to discretize the integral in Eq. 4 affects linearly the computational cost of our model. We trained a model on the Euler-quadrant dataset with $0, 2, 6, 14, 30, 62$ integration steps, 30 being the number of steps used in the rest of the paper. The validation loss after 50 epochs is shown on Tab. 4. On the one hand, the precision of the models is relatively stables w.r.t. the number of integration steps as long as it is not smaller than 6, suggesting the potential to reduce this number depending on the application, which could lead to computational savings. On the other hand, the case $n_{\text{steps}} = 0$ is closely related to the meta-learning approach proposed in (Wang et al., 2022) based on a discrete-time formulation, and its poorer performance highlights the benefit of the continuous-time formulation adopted in IC-NPDE.

Table 4: Effect of the number of intermediate discretization steps in the integration of the operator $f_\theta$ from $t$ to $t + 1$, on the Euler dataset. $n_{\text{steps}} = 0$ indicates that the operator is applied only once to obtain $u_{t+1}$ from $u_t$.

| $n_{\text{steps}}$ | 0 | 2 | 6 | 14 | 30 | 62 |
|---|---|---|---|---|---|---|
| **NRMSE** $t + 1$ | 0.080 | 0.068 | 0.045 | 0.045 | 0.045 | 0.045 |

**Single dataset training.** Training on multiple datasets simultaneously exposes the model to variations in the class of PDE, its coefficients, and initial conditions. We also evaluate models trained on individual datasets, which confronts our model to contexts with only the last two sources of variability. As shown in Tab. 5, our model remains competitive across most datasets. Comparing Tabs. 5 and 2, our model performs even better when trained on multiple datasets. It demonstrates that our hypernetwork, going beyond standard transformers, excels on data with significant variability, indicating improved generalization properties, crucial for multi-physics pretraining.

Table 5: Next-steps prediction performance for models trained separately on individual datasets.

| Dataset | Model | NRMSE | | | | |
|---|---|---|---|---|---|---|
| | | $t+1$ | $t+4$ | $t+8$ | $t+16$ | $t+32$ |
| Burgers | AViT | 0.0065 | 0.033 | 0.10 | 0.14 | **0.19** |
| | IC-NPDE (ours) | **0.0020** | **0.020** | **0.077** | 0.14 | 0.30 |
| Diffusion-reaction | AViT | 0.0020 | 0.061 | 0.19 | **0.28** | **0.56** |
| | IC-NPDE (ours) | **0.00039** | **0.060** | **0.16** | 0.37 | 0.73 |
| Navier-Stokes (incomp.) | AViT | 0.0042 | **0.015** | **0.041** | 0.090 | 0.29 |
| | IC-NPDE (ours) | **0.0040** | 0.018 | 0.045 | **0.075** | **0.28** |
| Shearflow (incomp.) | AViT | **0.010** | 0.11 | 0.55 | **0.29** | **1.1** |
| | IC-NPDE (ours) | 0.012 | **0.10** | **0.40** | 0.35 | 1.3 |
| Euler (comp.) | AViT | 0.046 | 0.075 | 0.20 | 0.34 | 0.76 |
| | IC-NPDE (ours) | **0.034** | 0.075 | **0.17** | **0.29** | **0.66** |

## 5 CONCLUSION

In this paper, we introduced in-context neural PDE (IC-NPDE), a general and efficient framework for in-context learning of dynamical systems governed by unknown temporal PDEs. Our approach integrates neural PDE solvers, which leverage continuous-time dynamics and spatial translation equivariance, with transformer-based hypernetworks that adapt to varying contexts to generate the solver parameters. Compared to standard in-context learning methods based purely on transformers, IC-NPDE achieves superior generalization and fine-tuning performance.

The integrated model in our framework, implemented using CNNs, is primarily inspired by finite difference schemes on a uniform mesh. However, many challenging problems in physics involve non-uniform meshes or arbitrary geometries. In such cases, adopting finite volume or finite element schemes could be achieved using graph neural networks instead, as demonstrated in recent works (Pfaff et al., 2020; Zhou et al., 2022; Brandstetter et al., 2022; Zhou et al., 2023). There are also some other promising directions for expanding the capabilities of IC-NPDE. First, The neural ODE-like structure enables flexible inclusion of future time step labels (e.g., $t+2$) in training objective, allowing adaptation to varying data evolution speeds. Additionally, while our method has been validated on time-independent dynamics, extending it to time-dependent systems requires incorporating temporal inputs into the hypernetworks, which is left for future exploration. Moreover, the integrated network in our framework, implemented using CNNs, is primarily inspired by finite difference schemes. Exploring other numerical methods, such as spectral methods, could lead to architectures similar to FNOs.

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

## A  DATASET DESCRIPTION

We provide the description of the datasets considered in this paper, along with their underlying PDEs. All the corresponding data is either publicly available (Takamoto et al., 2022) or is generated using public code following instructions below (Burns et al., 2020; Clawpack Development Team, 2021).

### A.1  BURGERS EQUATIONS

The Burgers equations model the evolution of a 1D viscous fluid. They combine a nonlinear advection term with a linear diffusion term

$$\partial_t u_t + u_t \partial_x u_t = \frac{\nu}{\pi} \partial_{xx}^2 u_t$$

where $\nu$ is the diffusion coefficient.

The boundary conditions are set to periodic. The dataset is part of PDEBench (Takamoto et al., 2022) and was generated using a temporally and spatially 2nd-order upwind difference scheme for the advection term, and a central difference scheme for the diffusion term. Refer to Figs. 4,9 for examples of trajectories.

## A.2 SHALLOW WATER EQUATIONS

Shallow-water equations, derived from Navier-Stokes equations present a suitable framework for modelling free-surface flow problems. They take the form of a system of hyperbolic PDEs

$$\partial_t h + \partial_x h + \partial_y hv = 0\,,$$

$$\partial_t hu + \partial_x (u^2 h + \frac{1}{2} g_r h^2) + \partial_y uvh = -g_r h \partial_x b\,,$$

$$\partial_t hv + \partial_y (v^2 h + \frac{1}{2} g_r h^2) + \partial_x uvh = -g_r h \partial_y b\,,$$

where $h$ is the water depth, $u, v$ are the velocities in horizontal and vertical direction, $b$ is the bathymetry field, and $g_r$ describes the gravitational acceleration.

The initial state is a circular bump in the center of the domain. The dataset is part of PDEBench (Takamoto et al., 2022) and was generated using a finite volume solver (Ketcheson et al., 2012). Refer to Fig. 4 for an example trajectory.

## A.3 DIFFUSION REACTION EQUATIONS

A 2D diffusion-reaction equation models how a substance spreads and reacts over time, capturing the combined effects of diffusion and chemical or biological reactions in two dimensions

$$\partial_t u = D_u \partial_{xx}^2 u + D_u \partial_{yy} u + R_u\,,$$

$$\partial_t v = D_v \partial_{xx}^2 v + D_v \partial_{yy} u + R_v\,,$$

where $D_u, D_v$ are the diffusion coefficients for the activator $u$ and inhibitor $v$ and $R_u, R_v$ are the respective reaction functions, which takes the form $R_u(u, v) = u - u^3 - k - v$ and $R_v(u, v) = u - v$ where $k$ is a constant. The initial states are random Gaussian white noises. The problem involves no-flow Neumann boundary condition, that is $D_u \partial_x u = 0, D_v \partial_x v = 0, D_u \partial_y u = 0, D_v \partial_y v = 0$ on the edges of the square domain. The dataset is part of PDEBench (Takamoto et al., 2022) and was generated using a finite volume method as spatial discretization and fourth-order Runge-Kutta method as time discretization. Refer to Figs. 8,9 for examples of trajectories.

## A.4 INCOMPRESSIBLE NAVIER-STOKES

This dataset considers a simplification of Navier-Stokes equation that writes

$$\nabla \cdot \mathbf{v} = 0 \ , \ \ \rho(\partial_t \mathbf{v} + \mathbf{v} \cdot \nabla \mathbf{v}) = -\nabla p + \eta \Delta \mathbf{v} + \mathbf{u}$$

where $\mathbf{v}$ is the velocity vector field, $\rho$ is the density and $\mathbf{u}$ is a forcing term and $\nu$ is a constant viscosity.

Initial states and forcing term $\mathbf{u}$ are each drawn from isotropic random fields with a certain power-law power-spectrum. The boundary conditions are Dirichlet, imposing the velocity field to be zero at the edges of the square domain. The dataset is part of PDEBench (Takamoto et al., 2022) and was generated using a differentiable PDE solver (Holl et al., 2020). Refer to Figs. 8,10 for examples of trajectories.

## A.5 SHEARFLOW

This phenomenon concerns two layers of fluid moving in parallel to each other in opposite directions, which leads to various instabilities and turbulence. It is governed by the following incompressible Navier-Stokes equation

$$\frac{\partial u}{\partial t} - \nu \Delta u + \nabla p = -u \cdot \nabla u\,.$$

where $\Delta = \nabla \cdot \nabla$ is the spatial Laplacian, with the additional constraints $\int p = 0$ (pressure gauge). In order to better visualize the shear, we consider a passive tracer field $s$ governed by the advection-diffusion equation

$$\frac{\partial s}{\partial t} - D\,\Delta s = -u \cdot \nabla s\,.$$

We also track the vorticity $\omega = \nabla \times u = \frac{\partial u_z}{\partial x} - \frac{\partial u_x}{\partial z}$ which measures the local spinning motion of the fluid. The shear is created by initializing the velocity $u$ at different layers of fluid moving in opposite horizontal directions. The fluid equations are parameterized by different viscosity $\nu$ and tracer diffusivity $D$.

The data was generated using an open-source spectral solver (Burns et al., 2020) with a script that is publicly available. Refer to Fig. 10 for an example trajectory.

## A.6 Euler equations (compressible)

Euler equations are a simplification of Navier-Stokes in the absence of viscosity

$$\partial_t U + \partial_x F(U) + \partial_y G(U) = 0\,,$$

with

$$U = \begin{pmatrix} \rho \\ \rho u \\ \rho v \\ e \end{pmatrix}\,,\ F = \begin{pmatrix} \rho u \\ \rho u^2 + p \\ \rho uv \\ u(e+p) \end{pmatrix}\,,\ G = \begin{pmatrix} \rho v \\ \rho uv \\ \rho v^2 + p \\ v(e+p) \end{pmatrix}$$

where $\rho$ is the density, $u, v$ are the horizontal and vertical velocities, $p$ is the pressure and $e$ is the energy defined by

$$e = \frac{p}{\gamma - 1} + \frac{1}{2}\rho(u^2 + v^2)\,.$$

The initial state is a piecewise constant signal composed of quadrants, which then evolves in multi-scale shocks.

The data has periodic boundary conditions and was generated using `CLAWPack` (Clawpack Development Team, 2021; Mandli et al., 2016), which is an open-source software for solving hyperbolic conservation laws, with a script that is publicly available. Refer to Fig. 10 for an example trajectory.

## A.7 Compressible Navier-Stokes

The compressible Navier-Stokes describe the motion of a fluid flow

$$\partial_t \rho + \nabla \cdot (\rho \mathbf{v}) = 0$$
$$\rho(\partial_t \mathbf{v} + \mathbf{v} \cdot \nabla \mathbf{v}) = -\nabla p + \eta\Delta\mathbf{v} + (\zeta + \eta/3)\nabla(\nabla \cdot \mathbf{v})\,,$$
$$\partial_t(\epsilon + \frac{1}{2}\rho v^2) + \nabla \cdot ((\epsilon + p + \frac{1}{2}\rho v^2)\mathbf{v} - \mathbf{v} \cdot \sigma') = 0$$

where $\rho$ is the density, $\mathbf{v}$ is the velocity vector field, $p$ the pressure, $\epsilon = p/(\Gamma - 1)$ is the internal energy, $\Gamma = 5/3$, $\sigma'$ is the viscous stress tensor, and $\eta, \zeta$ are the shear and bulk viscosity. The boundary conditions are periodic.

The dataset is part of PDEBench (Takamoto et al., 2022) and was generated using 2nd-order HLLC scheme (Toro et al., 1994) with the MUSCL method (Van Leer, 1979) for the inviscid part, and the central difference scheme for the viscous part. This dataset contains trajectories with only 5 steps into the future and was used solely for training.

## B BENCHMARK MODELS HYPERPARAMETERS

In this paper we compared our in-context neural PDE model with 4 baselines, a U-net (Ronneberger et al., 2015), a Fourier neural operator (Li et al., 2020) implemented using `neuralop` (Kovachki et al., 2023), an auto-regressive diffusion model (Kohl et al., 2024), and a AViT (McCabe et al., 2024) which is a transformer (Vaswani et al., 2017). We expose here the hyperparameters of these models used in the paper.

**U-Net.** We considered a standard U-Net (Ronneberger et al., 2015) with 4 down-(and up) sampling blocks, spatial filters of size 3 and initial dimension of 48. The resulting model has 17M learnable parameters.

**FNO.** We considered a standard Fourier neural operator (Li et al., 2020) with 4 Fourier blocks, spectral filters of size 16 (number of Fourier modes), and 128 hidden dimensions. The resulting model has 19M learnable parameters.

**AR-diffusion.** We considered an auto-regressive diffusion model (Kohl et al., 2024) based on a U-Net denoiser having 3 down-(and up) sampling blocks, with spatial filters of size 7 and 128 hidden dimensions. At inference, the generative process employs 100 diffusion steps. The resulting model has 7M learnable parameters.

**AViT.** We considered an axial vision transformer (McCabe et al., 2024) with patch size of 16, 12 attention layers, 12 attention heads per layer, 768 hidden dimensions. The resulting model has 158M learnable parameters.

## C TRAINING DETAILS

**Hyperparameters.** For joint training on multiple datasets, we considered the same hyperparameters than McCabe et al. (2024), with batch size of 8, gradient accumulation every 5 batches, epoch size of 2000 batches. For single model trainings, we considered batch size of 32 with no gradient accumulation. In certain cases where the optimization was unstable, in particular, when we tried using only 2 intermediate number of steps (see Tab. 6), we used gradient clipping, clipping the total norm of the gradients to a default norm of 1.0. The data is split into train, validation, and test sets with an 80%, 10%, and 10% division, respectively.

**Optimization.** All trainings were performed using the adaptive Nesterov optimizer (Xie et al., 2024) and a cosine schedule for the learning rate. Using AdamW optimizer with varying learning rate did not improve overall performance in the cases we tested. Both the single-dataset and multi-dataset experiments are run for a fixed number of 300 epochs. This means that during single-dataset training, the model sees 7 times more data from that dataset compared to a model trained on 7 datasets simultaneously, where each dataset is sampled less frequently. We used a weight decay of 0.001 and drop path of 0.1.

**Loss.** A normalized root mean square error is used for both monitoring the training of the model and assessing the performances in this paper. For two tensors $u$ (target) and $\hat{u}_{t+1}$ (prediction) with $C$ channels

$$\text{Loss}(u, \hat{u}) = \frac{1}{C} \sum_{c=1}^{C} \frac{\|u^c - \hat{u}^c\|_2}{\|u^c\|_2 + \epsilon}$$

where the $\ell^2$ norm $\|\cdot\|_2$ is averaged along space and $\epsilon$ is a small number added to prevent numerical instabilities. For a batch of data, this loss is simply averaged.

**Software.** The model trainings were conducted using python v3.11.7 and the PyTorch library v2.4.1 (Paszke et al., 2019).

**Hardware.** All model trainings were conducted using Distributed Data Parallel across 4 or 8 Nvidia H100-80Gb GPUs.

## D MORE EXPERIMENTAL RESULTS

**Additional rollout examples.** For models trained on multiple datasets jointly, on top of Fig. 4, additional rollouts are shown in Fig. 8. Refer to Tab. 2 for averaged metrics. For models trained on each dataset separately, Figs. 9,10, provide examples of rollouts. Refer to Tab. 5 for averaged metrics.

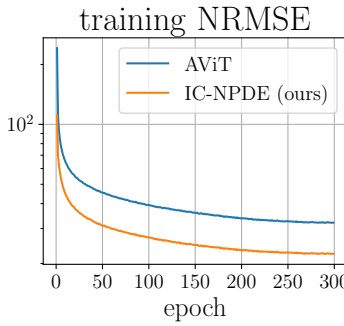
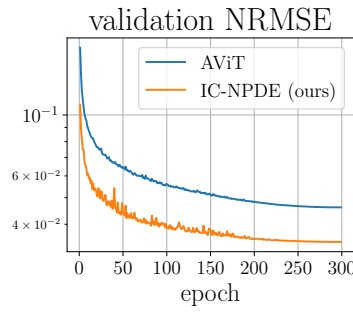

Figure 7: Training curve of AViT and IC-NPDE on Euler (compressible) 2D dataset.

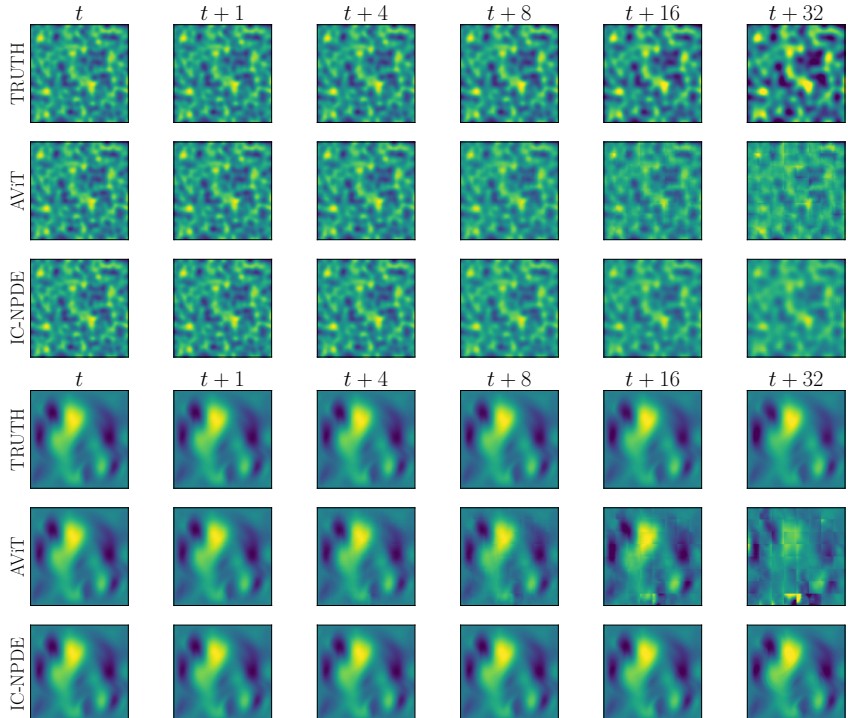

Figure 8: Examples of rollout trajectories from diffusion-reaction and incompressible Navier-Stokes datasets for models trained jointly on multiple datasets.

**Model size.** In our framework (see Eq. 4), both the hypernetwork size can be increased, to improve the estimation of the parameters $\theta$, and the operator size $f_\theta$, to provide more complex operators. The size $d_2$ of the hypernetwork $\psi_\alpha$ depends on the following hyperparameters, the number of attention layers, the number of attention heads in each layer, and the hidden dimension, which are set to 12, 6, 384, respectively, in the paper. We considered two other sets of hyperparameters 4, 3, 192 and 12, 12, 768. The size $d_1$ of the operator $f_\theta$ depends on the hidden dimension $C_{\text{hidden}}$, which is set to 64 in the paper. We considered $C_{\text{hidden}} = 32$ and $C_{\text{hidden}} = 128$. Tab. 6 shows the sizes for these hyperparameter choices, as well as the associated accuracy on the Euler dataset after 50 epochs. As we can see, the main performance improvements can be achieved by increasing the class of integrated network. However, training a larger hypernetwork on a single dataset appears to be more challenging. Note that according to the naive scaling specified in the third row of Tab. 6, the final MLP in the hypernetwork becomes quite large, as it takes an input with 768 channels and needs to output $40k$ channels, which is the dimension of $\theta$. A more refined strategy should be employed to scale the hypernetwork $\psi$.

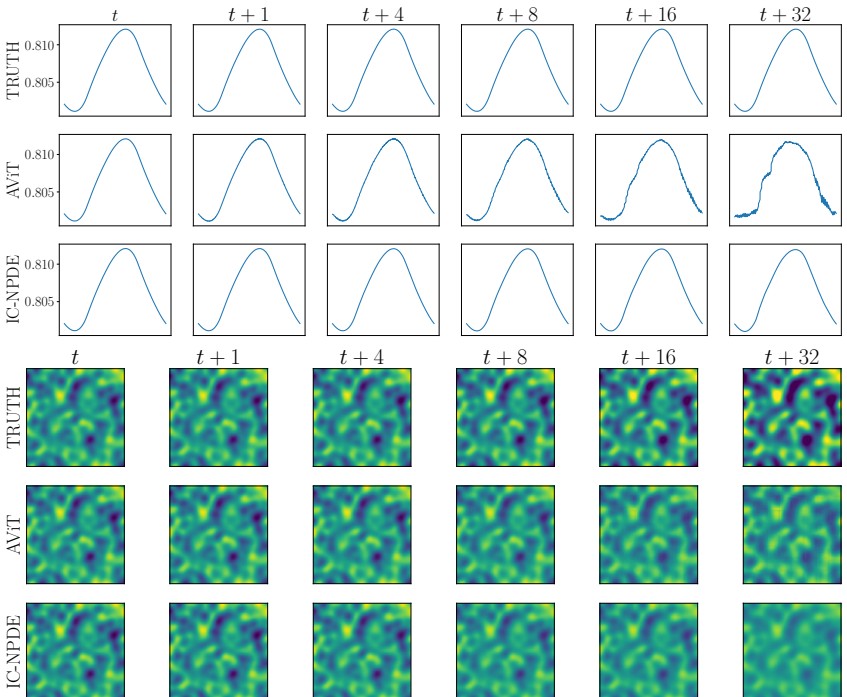

Figure 9: Examples of rollout trajectory from Burgers 1D and diffusion-reaction datasets, for models trained on each of these datasets separately.

Table 6: Influence of the model size on the accuracy, on the Euler dataset. Each table corresponds respectively to variations over $\alpha$ and $\theta$. Gray rows indicate the default values chosen in the paper.

| hyperparameters | | | num. weights $\alpha$ | NRMSE |
|---|---|---|---|---|
| num. layers | num. heads | dim | | |
| 4 | 3 | 192 | 11m | 0.046 |
| 12 | 6 | 384 | 55m | 0.046 |
| 12 | 12 | 768 | 189m | 0.048 |
| hyperparameter $C_{\text{hidden}}$ | | | num. parameters $\theta$ | NRMSE |
| 32 | | | 11k | 0.054 |
| 64 | | | 40k | 0.046 |
| 128 | | | 149k | 0.043 |

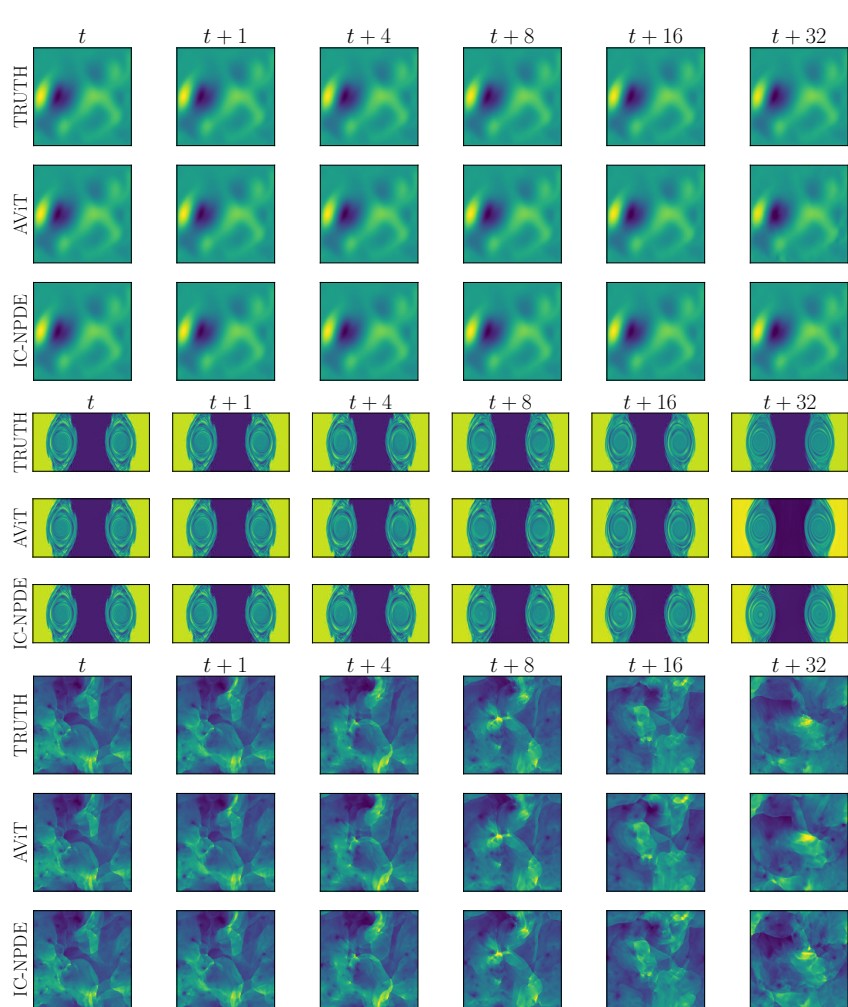

Figure 10: Examples of rollout trajectory from Navier-Stockes (incompressible), shearflow (incompressible) and Euler (compressible) datasets, for models trained on each of these dataset separately.

