# OpenReview forum: "In-Context Neural PDE: Learning to Adapt a Neural Solver to Different Physics"
_ICLR.cc/2025/Conference — Submitted to ICLR 2025_

### Official Review · Reviewer_PDbc · 2024-10-30

**Soundness:** 4
**Presentation:** 4
**Contribution:** 1
**Rating:** 3
**Confidence:** 4

**Summary:**

This paper proposes the IC-NPDE framework for predicting future states in dynamical systems governed by unknown PDEs. Leveraging In-Context Learning (ICL), it employs a transformer-based hypernetwork to estimate parameters for neural ODEs, which use CNNs as vector fields. Experimental results demonstrate that IC-NPDE can operate effectively across multiple datasets, with the information bottleneck enhancing its generalization performance.

**Strengths:**

- The writing in this paper is good. The logic is clear, and the background and preliminaries are well-explained, making it highly enjoyable to read.
- The motivation is thoroughly articulated, effectively establishing the relevance and importance of the proposed IC-NPDE framework. This well-defined motivation makes the contributions and experimental results both meaningful and easy to appreciate.
- A significant strength of IC-NPDE is its ability to be directly trained on multiple different PDEs, showcasing its versatility and robustness. This capacity for multi-PDE training expands its applicability in real-world scenarios where multiple physical processes often interact.

**Weaknesses:**

- A notable limitation of the paper is the limited contribution and novelty. Generally speaking, the key techniques utilized in this work are already well-studied and very common in the community of modeling dynamical system. Examples include Neural PDEs [1, 2], using NNs to predict the parameters of ODEs [3], and context-based learning for dynamical systems [4].
- Although some related works may have different settings, this paper would benefit from expanding the related work section to provide a more thorough review of existing methodologies. Specifically:
  - Other Neural PDE Approaches: including [1, 2, 6]
  - Multiple Environment Settings: including [4, 5]. They should also be used as baselines in Section 4.4, single dataset training.
  - Predicting ODE Parameters with Neural Networks: [3]
- In lines 126-127, the author mentioned that the spatial dimension n ranges from 1 to 3; however, all experiments in the paper are limited to 1D and 2D PDEs, with no inclusion of 3D PDEs. Extending the experiments to cover 3D and even higher-dimensional PDEs would provide a more comprehensive evaluation of IC-NPDE’s scalability and performance across spatial complexities.
- All experiments in the paper are conducted on simulated data, with no evaluation on real-world applications. Maybe climate prediction in different areas can be a good choice.
- The experimental evaluation would be strengthened by including additional baseline models, even if multi-PDE settings are less common. Baselines that focus on learning individual PDEs or on learning within the same PDE with varying parameters should be incorporated to provide a more comprehensive benchmark in Table 3 and Table 5, respectively.

**References**:

[1] Dulny, Andrzej, Andreas Hotho, and Anna Krause. "NeuralPDE: modelling dynamical systems from data." German Conference on Artificial Intelligence (Künstliche Intelligenz). Cham: Springer International Publishing, 2022.

[2] Gelbrecht, Maximilian, Niklas Boers, and Jürgen Kurths. "Neural partial differential equations for chaotic systems." New Journal of Physics 23.4 (2021): 043005.

[3] Linial, Ori, et al. "Generative ode modeling with known unknowns." *Proceedings of the Conference on Health, Inference, and Learning*. 2021.

[4] Kirchmeyer, Matthieu, et al. "Generalizing to new physical systems via context-informed dynamics model." International Conference on Machine Learning. PMLR, 2022.

[5] Yin, Yuan, et al. "LEADS: Learning dynamical systems that generalize across environments." Advances in Neural Information Processing Systems 34 (2021): 7561-7573.

[6] Verma, Yogesh, Markus Heinonen, and Vikas Garg. "Climode: Climate and weather forecasting with physics-informed neural odes." *arXiv preprint arXiv:2404.10024* (2024).

**Questions:**

- The expression "much smaller neural PDE solver" in lines 66-67  is a little weird. IC-NPDEs are based on the Neural ODE architecture. Typically, in Neural ODEs, the neural network defines the vector field, while standard numerical ODE solvers perform the integration.
- The authors mention in lines 221–222 that Transformers are used as hypernetworks due to their auto-regressive properties. However, original Transformers with softmax attention are not inherently auto-regressive; it is only with modifications, such as those in [7], that Transformers adopt an auto-regressive structure.
- I am a bit unclear on why the authors chose RK4 as the ODE solver, given that it is a fixed-step-size solver. Adaptive-step-size solvers like Dopri5 are generally more common in the Neural ODE community. With an adaptive solver, the ablation study on step size might be unnecessary.

**References**:

[7] Katharopoulos, Angelos, et al. "Transformers are rnns: Fast autoregressive transformers with linear attention." *International conference on machine learning*. PMLR, 2020.

---

> ### Author Response · Authors · 2024-11-24
>
> We thank the reviewer for their thorough examination of our work and for the useful suggestions.
>
> - On the very important point of the contribution, we refer the reader to the general comment which explains in detail why there are very few baselines currently capable of tackling the task we address in our paper.
> Regarding the references you mention, [1,2] assume the PDE class and the PDE coefficients are fixed, [3] builds a class of operators $f_\theta$ that is tested in an ODE setting for only one class of ODE, while our approach defines a class of operators tested in a setting that encompasses multiple PDE classes, [4] assumes that we know whether the context is generated from a different PDE class and PDE coefficients.
> - We thank you for the suggestions for complementary baselines. Note, that they make additional assumptions on the contexts.
> - This will be corrected in the new version. We did not include coarser results in addition to the 1024, 128x128 and 128x256 resolutions already presented in the paper. However, it is worth noting that experiments conducted at a 512x512 resolution demonstrate the same improved sample efficiency and final accuracy.
> - We indeed plan to test our approach on real-world observational data, with climate prediction being an interesting candidate.
> - The baselines included in Table 3. (Unet, FNO, AR-diffusion) are known to be the main competitive methods in the case of a single PDE, but we will consider some of your aforementioned suggestions in the next version of the paper.
>
> Answers to your questions.
> - We will clarify this point in the next version of the paper.
> - We will also clarify this point in the next version of the paper.
> - We originally ran experiments with the Dopri5 solver, but these appeared to be significantly slower than a RK4 solver, without necessarily improving performance.

---

> > ### Comment · Reviewer_PDbc · 2024-11-26
> >
> > 1. While other baselines may involve more assumptions, I believe it is still important to compare against them under their respective conditions, as this would provide a clearer and more comprehensive demonstration of performance.
> >
> > 2. My concerns remain entirely unaddressed. The authors need to put in more effort to adequately address these issues. If no modifications or additional experiments are presented during the discussion, I am inclined to maintain or potentially lower my current score.

---

> > ### Comment · Reviewer_PDbc · 2024-11-26
> >
> > By the way, I don’t believe I raised any questions regarding sampling resolution. Could you please provide a more detailed explanation as to why 3D systems were not included?

---

> > > ### Author Response · Authors · 2024-12-02
> > >
> > > We did not originally include 3D systems since the most important baseline in our paper is based solely on 1D and 2D data [8]. Furthermore, there are very few studies addressing this task in 2D, and, to the best of our knowledge, none that tackle 3D. Given the challenges inherent to this task, we focused on developing and refining our approach in lower dimensions before considering 3D extensions.
> > >
> > > [8] McCabe et al., Multiple Physics Pretraining for Physical Surrogate Models. 2024.

---

> ### Comment · Reviewer_PDbc · 2024-11-29
>
> Since my concerns remain entirely unaddressed, and currently the authors do not address them totally, I decide to lower my current score from 5 to 3.

---

> > ### Author Response · Authors · 2024-12-02
> >
> > We are preparing a revised version of the paper with improved clarity and additional experiments, which will be submitted to another venue.
> >
> > We have worked to address the points you raised. While we are unsure what in our response justifies lowering your score, we thank the reviewer for their feedback and suggestions.

---

> ### Comment · Reviewer_PDbc · 2024-12-02
>
> 1. I understand that the primary assumption of your work focuses on multiple PDEs. However, several experiments do not entirely align with this focus (e.g., single-dataset training). In such cases, baselines other than AViT are also necessary. Unfortunately, throughout the entire discussion period, the authors failed to address these alternative baselines.
>
> 2. 3D PDEs and real-world applications are still not included in the paper.
>
> 3. There is no revision for the paper. All the raised questions are not included in the paper during the discussion period.
>
> For the ICLR rebuttal, the authors had the opportunity to revise the paper and add more experiments to convince reviewers. However, during the entire discussion period, they neither revised the paper nor added additional experiments. Their only response was, "We are working on that." I find this lack of engagement disrespectful to us as reviewers. To be frank, I believe most authors of other papers would have already revised their papers by now, given that the modification deadline has passed. I am deeply disappointed, as we have dedicated significant effort to help improve the quality of this work, yet no progress has been made.
>
> Until I see meaningful improvements to this paper (e.g., new experiments or revisions), I will maintain my current score as a form of protest against the authors' attitude.

---

> ### Author Response · Authors · 2024-12-03
>
> Dear reviewer,
>
> We thank you once again for your feedback and suggestions, which, rest assured, will be taken into account in the next version of the paper.

---

### Official Review · Reviewer_JnHg · 2024-10-30

**Soundness:** 3
**Presentation:** 3
**Contribution:** 3
**Rating:** 5
**Confidence:** 1

**Summary:**

This paper presents a novel method, in-context neural PDE, for predicting the next state of a dynamical system governed by unknown PDEs. The approach employs a large transformer-based hypernetwork to process successive states and generate parameters for a smaller neural ODE-like solver, which then predicts the next state via time integration. Numerical experiments validate the effectiveness of the proposed method.

**Strengths:**

(1) Combining the ICL approach with differentiable PDE solvers for spatiotemporal prediction is a promising direction.

(2) The paper is well-written, making it easy to read and understand.

**Weaknesses:**

(1)  The selection of numerical schemes should be more diverse.

(2)  Since the proposed method involves ICL and numerical schemes, what is the computational complexity? Is the model sufficiently robust? The paper lacks relevant explanations on these aspects.

**Questions:**

Refer to weakness.

---

> ### Author Response · Authors · 2024-11-15
> **Asking for precisions**
>
> Dear reviewer,
>
> We thank you for your comments on our paper, but we need precisions to address them accurately.
> - "The selection of numerical schemes should be more diverse" \
> In the paper, we rely on a Runge-Kutta 4 numerical scheme, which is widely used for solving PDEs and well implemented in python. Are you suggesting using simpler or higher order numerical schemes? Are you suggesting adapting the numerical scheme based on the dataset?
> - "Is the model sufficiently robust?" \
> Could you precise the type of robustness you are referring to? (e.g. robustness to noise in the data, to outliers, to adversarial attacks ...).

---

> ### Author Response · Authors · 2024-11-24
>
> We thank the reviewer again for their comments on our paper. We would appreciate the opportunity to address the reviewer's points in detail once we have further clarification on their feedback.

---

### Official Review · Reviewer_f5UB · 2024-11-01

**Soundness:** 2
**Presentation:** 3
**Contribution:** 2
**Rating:** 3
**Confidence:** 4

**Summary:**

This work tackles the task of spatiotemporal forecasting for PDEs governed by different unknown physics; each trajectory is thus described by a specific equation or set of coefficients. To build a neural solver able to generalize to trajectories described by different physics, the authors advocate the use of in-context learning to adapt a cheap neural ODE solver via a transformed-based hypernetwork. The hypernetwork uses preceding T successive states for adapting the forecaster network to each unknown physics. The method is evaluated on a wide range of datasets and performs better or is competitive with existing baselines. It is also has been adapted to new physics via fine-tuning.

**Strengths:**

The paper is well written and easy to follow throughout and the motivations well explained.

The tackled task is important. Most of the times, machine learning approaches for solving PDE consider that a large number of trajectories are available for one single PDE equation with fixed parameters. In practical scenarios, multiple and unknown physics are expected, leading to very different dynamical behaviors, which need to be captured by a single neural network.

The forecaster network is more interpretable than existing data-driven approaches via a neural ODE using a simple CNN. The authors connect it to classical numerical methods.

**Weaknesses:**

Novelty:
- the technical contribution is relatively low. Adapting neural ODE-solver to different physics has already been explored. The use of hypernetworks for adapting neural networks has been well studied in the meta-learning litterature. it has also been used for dynamical systems to condition a neural-ode like solver [1]. The differences with [1] is 1) that environments are supposed known, each environment describing a specific physic, while here, the environments would correspond to the first T states. 2) the use of ICL against weights updates.
- icl works on quantized tokens in text, its use for continuous vectors and physical data is not straightforward. This has not been really well introduced in the paper. The authors notably justify their approach is close to [2], which employs a transformer approach for learning multiple physics using past states as input. It does not imply that [2] has ICL properties to me, as stated in the paper.

Evaluation:
- empirical results: while the method is competitive to AViT, it should have been compared to more related works. The method is presented as a meta-learning framework and comparison with respect to these approaches are thus important and should be included, to justify the strength of ICL compared to existing meta-learning strategies [1, 3, 4, 5, 6].

Choice of title:
- the title is not really appropriate: It sounds like the network is able to adapt to new physics (new equations) without fine-tuning, only by explicitly giving a new context, but the network needs to be fine-tuned as some layers of the network are specific to each dataset. I don't really understand why the authors decided to train a network on different PDEs simultaneously for leveraging the capacities of ICL, if fine-tuning is still needed for unseen PDEs.

[1] CoDA - Kirchmeyer et al., Generalizing to New Physical Systems via Context-Informed Dynamics Model. ICML, 2022.

[2] MPP - McCabe et al., Multiple Physics Pretraining for Physical Surrogate Models. 2024, https://arxiv.org/pdf/2310.02994.

[3] DyAd - Wang et al. 2022, Meta-Learning Dynamics Forecasting Using Task Inference, NeurIPS, 2022.

[4] CAVIA - Zintgraf et al., Fast Context Adaptation via Meta-Learning. ICML, 2019.

[5] CAMEL - Blanke et al., Interpretable Meta-Learning of Physical Systems, ICLR, 2024.

[6] FOCA - Park et al., First-order Context-based Adaptation for Generalizing to New Dynamical Systems, 2023, https://openreview.net/pdf?id=AW0i0lOhzqJ.

**Questions:**

To me, when I think of ICL in text, arbitrary sized examples can be given to adapt the language model to a new task. Can your method be applied to arbitrary sized sequences, i.e., can the hypernetwork adapt the integrated network given N past states, where N can vary at inference ?

In the umap visualisation, at epoch 0, the points seem to be already very well clustered. How do you explain that?

I am bit concerned with the choices of the datasets. First, qualitatively, it is hard to see very big changes from t=0 to t=32. But more importantly, the designed task is to adapt the network to different unknown physics. It is not clearly stated in the paper. How much different physics are present ? Are the pde coefficients used during training the same during inference or test is done on new coefficients?

Concerning the NRMSE results, you presented results at specific time-steps. Do the results correspond to NRMSE for that specic frame at time T or for average score from time 0 to T? It would have been nice to include an average score over the full trajectory.

---

> ### Author Response · Authors · 2024-11-24
>
> We thank the reviewer for their thorough examination of our work and their useful suggestions.
>
> - "the technical contribution is relatively low." \
> We strongly disagree on this point, and it is actually reflected by comparing our approach with [1] (which we now do in the paper).
> Being able to predict a PDE when it is unknown is crucial in many applications, for example, because the exact PDE may be difficult to estimate or derive.
> Such applications can be addressed with our method but cannot be tackled with [1], since [1] assumes the underlying PDE is known in order to use the appropriate pretrained common weights, denoted as $\theta_c$ in their paper.
> For the generic task tackled in our paper (which is clarified in the general comment),
> we cannot rely on pre-identified weights for known PDEs and must build a large network to predict the parameters from the context. This has never been done in the literature, and our paper demonstrates that it not only works but also outperforms approaches that do not explicitly leverage a numerical integrator, such as [2].
>
> - "icl works on quantized tokens" \
> We refer the reviewer to the general comment on the use of the term "in-context learning".
>
> - "empirical results:" \
> While we agree that more comparisons would be beneficial, there are actually very few baselines that offer a fair comparison. We refer the reviewer to the general comment regarding the task and the novelty of our approach. Specifically, [1] assumes that we know whether the context is generated from a different PDE class and PDE coefficients; [3] is trained on a single PDE class (e.g., turbulent Navier-Stokes, surface temperature, or ocean currents); [4] proposes to learn shared parameters across related tasks, similar to [1], requiring knowledge of the underlying PDE class for a context; [5] assumes that the PDE class is known and also assumes linearity in the parameters to predict on contexts with different PDE coefficients; and [6] also assumes that the PDE class is known in order to learn a unique set of shared parameters.
>
> - "The title is not really appropriate" \
> This is an important point. We decided to train a network on different PDEs simultaneously because we had no choice: the task involves learning from multi-physics contexts with a variety of PDEs without knowing the exact PDE.
> Our method would indeed require fine-tuning on unseen classes of PDEs; however, the layers that need to be trained for each dataset are minimal—the very first layer of the hypernetwork (since the context may have different input channels) and the very last layer of the hypernetwork (because the number of parameters of the predicted operator slightly varies for the same reason). As a consequence, the majority of the weights in our hypernetwork are shared across all contexts.
> In contrast, [1,3], for example, fine-tune a new set of parameters every time the set of PDE coefficients is changed, even of a small amount.
>
> Answer to your questions
> - To be able to change the context length at inference, with minimal loss of performance, we should train on varying context lengths, which we didn't do in the paper.
> - We believe the initial clustering in the UMAP space is due to the presence of different types of initial conditions in the dataset, as described in the PDEBench paper [7] (Eq. 17). To achieve the cleanest visualization, we should consider using a dataset with a fixed set of initial conditions drawn from the same distribution, each evolved with a different time operator corresponding to distinct PDE coefficients $\eta$. Such datasets are hard to find, and we opted to rely on the standard PDEBench [7] dataset.
> - We are open to any suggestions for more relevant datasets that would be considered convincing and pertinent in the literature. In our paper, the PDE coefficients for most of the datasets we used are available in the PDEBench paper [7]. Each PDE class includes between 4 and 30 PDE coefficients values. For example, the shearflow and Euler datasets (see Table.5) consist of 28 and 20 PDE coefficient values, respectively, with 40 and 500 different initial conditions, and 200-timestep-long trajectories per coefficient for both datasets.
> When evaluating the model in Tables 2 and 5, we include the same PDE coefficients in the validation set as in the training set.
> This important information will be mentioned in the next version of the paper.
> - The NRMSE is computed at a specific time-step, averaged over enough trajectories to ensure that variance is negligible.
>
> [7] PDEBench - An Extensive Benchmark for Scientific Machine Learning, NIPS, 2022

---

> > ### Comment · Reviewer_f5UB · 2024-11-26
> >
> > I would like to thanks the authors for their remarks and comments.
> >
> > Regarding the comparison with [1], I agree that the assumptions made with your work are different. In [1], it is assumed that the trajectories are part of environments, thus some form of knowledge is known for the PDE class or coefficients. But, I still think that your work could be compared to these meta-learning works for solving dynamical systems (e.g., the ones I cited in my first response), as these works could be adapted to your setting and adaptation could be done on the first frames of each trajectory, instead of assuming knowledge of environments. This could strengthen the paper in my opinion.
> >
> > Regarding the in-context learning concerns, I agree with you that your work does not possess standard 'in-context' properties. I did not mean that in-context learning could not work in another setting, as other works explored alternatives for PDEs [1, 2], but I would have appreciated that the proposed strategy for doing in-context was a bit more detailed and connected to existing works in different fields (e.g. NLP).
> >
> > I agree with the authors that only small number of parameters need to be tuned to adapt to new PDE classes, but it still requires weight updates like [3]. The term "different physics" in my opinion suggest that it could directly adapt to new PDE classes with in-context learning. It could be reformulated in a clearer manner maybe.
> >
> > Regarding the datasets, I think [3] generated different PDE that are publicly available on Huggingface, with changes in the initial conditions or PDE classes.
> >
> > I did not see any modifications nor added experiments yet in the paper if I am correct. I would maintain my score until an updated version is available, with my concerns addressed.
> >
> > [1] Chen et al., Data-Efficient Operator Learning via Unsupervised Pretraining and In-Context Learning, NeurIPS, 2024.
> > [2] Yang et al., In-Context Operator Learning with Data Prompts for Differential Equation Problems, PNAS, 2023.
> > [3] Herde et al., Poseidon: Efficient Foundation Models for PDEs, NeurIPS, 2024.

---

> > > ### Author Response · Authors · 2024-12-02
> > >
> > > We thank the reviewer for their feedback, particularly their relevant suggestions regarding datasets and experiments.
> > >
> > > We are preparing a revised version with enhanced clarity and additional experiments, which will be submitted to another venue.

---

### Official Review · Reviewer_eN6a · 2024-11-02

**Soundness:** 2
**Presentation:** 3
**Contribution:** 2
**Rating:** 3
**Confidence:** 4

**Summary:**

The paper introduces the "in-context neural PDE" framework, which decouples parameter estimation from state prediction for improved interpretability and generalization in predicting dynamic systems governed by unknown PDEs.

**Strengths:**

1. The research tackles a problem of high practical importance, which has significant potential for real-world applications in computational physics and engineering.
2. The paper is overall well written and easy to read.

**Weaknesses:**

1. The paper's related work section does not fully address similar settings explored in other articles, such as [1, 2], which use contrastive learning techniques to extract physical information from unknown PDE sequences for prediction. Furthermore, the authors claim that methods like [3, 4] lack inductive bias for physics and require a large amount of data to generalize, but the paper does not provide experiments to substantiate this claim.

2. The paper refers to "inductive bias for physical systems," but it lacks a clear definition. It's not evident how this bias is integrated into the model or how it influences the predictions for various physical systems. Does the term "inductive bias" solely refer to the use of rollouts with a small step size?

3. The paper claims its work is applicable to "DIFFERENT PHYSICS," but it does not sufficiently emphasize the specific physical quantities that vary across different PDEs, such as parameters, external forces, or boundary conditions, in the dataset descriptions (Table 1).

4. The experimental section only compares the proposed method with AViT, lacking comparisons with other baselines, particularly those that are also suitable for multi-physics scenarios.

5. The authors only demonstrate that the extracted representations have physical significance in the experiment shown in Figure 5, which is relatively straightforward, as there is a noticeable difference between fluids with viscosity coefficients of 0.1 and 0.01. It would be more compelling if the authors validated the physical meaning of their extracted representations in more challenging and diverse settings, similar to the studies referenced in articles [1, 2].

[1] Mialon G, Garrido Q, Lawrence H, et al. Self-supervised learning with lie symmetries for partial differential equations[C]. Advances in Neural Information Processing Systems, 2023, 36: 28973-29004.

[2] Zhang R, Meng Q, Ma Z M. Deciphering and integrating invariants for neural operator learning with various physical mechanisms[J]. National Science Review, 2024, 11(4): nwad336.

[3] McCabe M, Blancard B R S, Parker L H, et al. Multiple physics pretraining for physical surrogate models[C]. Advances in Neural Information Processing Systems, 2023.

[4] Yang L, Liu S, Meng T, et al. In-context operator learning with data prompts for differential equation problems[J]. Proceedings of the National Academy of Sciences, 2023, 120(39): e2310142120.

**Questions:**

Please see the weaknesses part.

---

> ### Author Response · Authors · 2024-11-24
>
> We thank the reviewer for their thorough examination of our work and for the useful comments.
>
> 1. We will incorporate [1,2] in the new version of the "related work" section. Regarding your last point, our paper does actually provide an experiment (Fig. 3), showing that a learned transformer [3] does not incorporate a translation-equivariant inductive bias. Such inductive bias is however present in the PDE solution as explained in our paper (l.160).
> 2. The term "inductive bias for physical systems" is indeed vague, and will be clarified in the new version. In particular, we refer to the translation-equivariance of the PDE solution (if the initial condition is translated, then the whole solution should be translated),
> as well as the fact that the function $g$, defining the PDE: $\partial_t u_t(x) = g(u_t(x),\partial_{x_1}u_t(x),\ldots)$ (l.124), is a pointwise function, in the sense that it operates the same way independently on the spatial position $x$.
> Our model incorporates such inductive biases by defining an operator $f_\theta$ which is a particular CNN, composed of a first convolution with kernels of size 5 (analogous to discretized differential operators), followed by 1x1 convolutions (effectively learning a pointwise function recombining these operators).
> 3. This will be precised in the new version.
> 4. This important point is addressed in the general comment.
> 5. We thank the reviewer for these suggestions. We indeed intend to showcase more results on the physical meaning of our methods in the next version of the paper.

---

> > ### Comment · Reviewer_eN6a · 2024-11-25
> >
> > Thank you for responding to my questions. As far as I'm aware, the author can upload a new version in ICLR 2025, while I haven't seen an updated paper. Hence, I decided to maintain my score.

---

> > > ### Author Response · Authors · 2024-12-02
> > >
> > > We thank the reviewer for their feedback and wish to inform them that we are working on a revised version of the paper, which will be submitted to another venue. This revision aims to improve clarity and include additional experiments.

---

### Official Review · Reviewer_LWEz · 2024-11-02

**Soundness:** 2
**Presentation:** 3
**Contribution:** 2
**Rating:** 3
**Confidence:** 5

**Summary:**

This paper introduces an in-context neural PDE framework that leverages a transformer-based hypernetwork to generate parameters for a neural ODE-like solver, effectively predicting the next state of dynamical systems governed by unknown temporal PDEs with limited time-lapse data. The authors claim that this approach decouples parameter estimation from state prediction, enhancing interpretability and outperforming standard transformer-based models in terms of generalization.

**Strengths:**

+ The proposed idea is straightforward.
+ The paper is well presented and easy to follow.
+ Results show the effectiveness of the model based on specified experimental settings.

**Weaknesses:**

- The major weakness lies in novelty. The model simply results from the stacking of well-known NN modules in the context of numerical integration. I don’t see any novelty compared with many other models reported recently in the literature.
- The examples considered to verify the efficacy of the model are too simple. The performance of the model should be at least tested on 3D datasets (e.g., 3D NS, 3D RD, etc.) to support the current conclusion. In addition, spatiotemporally coarser datasets should be considered.
- The baseline models are out of date. More recent neural PDE solvers (e.g., pre-trained PDE solvers) should be compared.
- The prediction horizon considered in the paper is too short (e.g., 32 steps). The model is only useful only long-term rollout prediction with high accuracy is retained.
- The literature review section is generally weak.

**Questions:**

Please see the weaknesses above.

---

> ### Author Response · Authors · 2024-11-15
> **Asking for precisions**
>
> Dear reviewer,
>
> We thank you for your comments on our paper, but we need precisions to address them accurately.
>
> Our paper addresses the task of predicting the next states of a spatio-temporal physical system, given only the previous $T$ states, across various (i) PDE classes, (ii) PDE coefficients, and (iii) initial conditions -- all without knowledge of the governing PDE.
>
> Few models can tackle this challenging task, as most assume known PDEs and coefficients (e.g., standard solvers) or single PDE with fixed coefficients (e.g., operator learning).
>
> - "I don't see any novelty compared with many other models reported in the literature" \
> Could you provide specific examples of the models you have in mind? While we are not claiming novelty in the individual transformer or neuralPDE components, the combination of the two -- using the transformer to predict the parameters of a neuralPDE solver -- is, to the best of our knowledge, novel.
> - "the baseline models are out of date" \
> Do you also include "AViT" in your statement? This baseline was made public a year ago, and is one of the first models proposed to tackle the above task.

---

> ### Author Response · Authors · 2024-11-24
>
> We thank the reviewer again for their comments and suggestions. Below are additional details supplementing our previous response.
>
> - "The major weakness lies in novelty." \
> We refer the reviewer to the general comment, which specifically addresses this point.
>
> - "The examples considered to verify the efficacy of the model are too simple" \
> We did not include coarser results in addition to the 1024, 128x128 and 128x256 resolutions already presented in the paper. However, it is worth noting that experiments conducted at a 512x512 resolution on datasets used in this paper (incomp. Navier-Stokes 512x512, comp. Navier-Stokes 512x512, Euler 512x512) demonstrate the same improved sample efficiency and accuracy.
>
> - "The baseline models are out of date." \
> This important point is addressed in the general comment.
>
> - "The prediction horizon considered in the paper is too short (e.g., 32 steps)" \
> As a matter of fact, a trained transformer (e.g. AViT [1]), which is the main baseline for our task, does struggle to predict correctly up to 32 steps already. Improvements in the rollout performance of our model are a step forward toward being able to predict longer horizons.
>
> [1] McCabe M, et al. Multiple physics pretraining for physical surrogate models. Advances in Neural Information Processing Systems, 2023.

---

> > ### Comment · Reviewer_LWEz · 2024-11-25
> > **Reply to authors's rebuttal**
> >
> > I thank the authors for the response. Unfortunately, none of my comments has been really addressed.
> >
> > - The novelty is not clarified. Stacking of well-known NN modules in the context of numerical integration does not represent sufficient novelty.
> >
> > - The examples tested are too simple to support the conclusion.
> >
> > - The baseline comparison is too weak.
> >
> > - A 32-step prediction is far from enough in long-term prediction.
> >
> > - The related work section needs comprehensive enhancement.
> >
> > I will be happy to increase the score once the above issues are seriously fixed.

---

> > > ### Author Response · Authors · 2024-12-02
> > >
> > > We thank the reviewer for their feedback. We are preparing a revised version of the paper to improve clarity and include additional experiments. This updated version will be submitted to another venue.

---

### Author Response · Authors · 2024-11-24

# General comment

We thank the reviewers for their thorough examination of our work and aim to demonstrate here why our approach is novel and constitutes a contribution to the literature.

## The task in our paper

We address a specific task in this paper, and we believe there have been some misunderstandings among the reviewers. We acknowledge that this confusion stems from a lack of clarity in the paper, which we will correct.

Our paper addresses the task of predicting the future states $u_{t+1},u_{t+2}\ldots$ of a spatio-temporal physical system, given only its past $T$ states $u_{t-T+1},\ldots,u_t$, across various (i) PDE classes, (ii) PDE coefficients, and (iii) initial conditions -- all without knowledge of the governing PDE.

This task has gained importance in recent months due to the community effort to build "foundation models" for PDEs, which must handle such variability. There are currently very few baselines for this task.


## Novelty of our approach

Given the above clarification, most models in the literature either assume that both (i) and (ii) are fixed [1,2] or that only (i) is fixed while knowing when a new context has different PDE coefficients (ii) [3,4]. Our model cannot make these assumptions. Consequently, comparisons with such models would be unfair.

However, we have included complementary comparisons with models that assume both (i) and (ii) are fixed, such as U-Net [6], FNO [7], and diffusion models [8], as shown in Table 3 of our paper. Our fine-tuned model compares favorably to these models.

While we agree that the paper does not make standalone contributions to a neural PDE framework or a transformer model, the combination of the two to tackle the aforementioned task is, to the best of our knowledge, novel. Compared to a pure transformer model such as [5], our approach combines the best of both worlds, achieving better performance and improved sample efficiency by utilizing a numerical solver.

## "In-Context" characterization of our approach

Solving the aforementioned task requires both estimating the underlying evolution operator and performing the prediction.
We described our method as "in-context" since the determination of the operator is done for each context (through the transformer).

Now, this leads to confusion because our approach does not possess standard 'in-context' properties (e.g., the ability to operate with multiple trajectories in the context) nor the ability to generalize to new datasets without fine-tuning.
However, it is important to note that a standard in-context learning approach, such as a recent work [9], made public after the submission of our paper, is not guaranteed to solve the task considered in our paper and may require fine-tuning to really generalize to unseen data.

[1] Dulny, Andrzej, Andreas Hotho, and Anna Krause. "NeuralPDE: modelling dynamical systems from data." German Conference on Artificial Intelligence (Künstliche Intelligenz). Cham: Springer International Publishing, 2022.

[2] Gelbrecht, Maximilian, Niklas Boers, and Jürgen Kurths. "Neural partial differential equations for chaotic systems." New Journal of Physics 23.4 (2021): 043005.

[3] Linial, Ori, et al. "Generative ode modeling with known unknowns." Proceedings of the Conference on Health, Inference, and Learning. 2021.

[4] Kirchmeyer, Matthieu, et al. "Generalizing to new physical systems via context-informed dynamics model." International Conference on Machine Learning. PMLR, 2022.

[5] McCabe et al., Multiple Physics Pretraining for Physical Surrogate Models. 2024.

[6] Ronneberger et al., U-net: Convolutional networks for biomedical image segmentation. 2015.

[7] Li et al., Fourier neural operator for parametric partial differential equations. 2020.

[8] Kohl et al., Benchmarking autoregressive conditional diffusion models for turbulent flow simulation. 2024.

[9] Serrano et al., Zebra: In-Context and Generative Pretraining for Solving Parametric PDEs. 2024.

---

### Meta-Review · Area_Chair_k1vq · 2024-12-19

**Metareview:**

This paper introduces a new in-context learning framework for PDE learning. The proposed approach can handle a range of PDE classes, coefficients, and initial conditions. This is an advantage over previously introduced in-context frameworks that typically focus on a single scenario. While this broader applicability is valuable, it remains somewhat incremental from a machine learning perspective.

The paper’s strengths lie in the simplicity and clarity of its presentation. However, all reviewers identified several weaknesses. Most critically, the framework was tested only on relatively simple tasks, and the authors did not compare its performance against state-of-the-art models. Moreover, there were concerns about the novelty of the proposed approach. Although the authors attempted to address some of these concerns, they did not provide a revised manuscript or new results.

In light of these issues, as reflected in the low review scores, I recommend rejecting this submission.

**Additional Comments On Reviewer Discussion:**

The authors attempted to address the reviewers concerns, however, several issues remained unanswered. The authors decided to prepare a revised version which will be submitted to another venue.

---

### Decision · Program_Chairs · 2025-01-22

Reject